# Residential Property Behavior Forecasting in the Metropolitan City of Milan: Socio-Economic Characteristics as Drivers of Residential Market Value Trends

**Marzia Morena** *, **Genny Cia**, **Liala Baiardi** and **Juan Sebastián Rodríguez Rojas**

Department of Architecture, Built Environment and Construction Engineering, Politecnico di Milano, Via Bonardi 9, 20133 Milan, Italy; genny.cia@polimi.it (G.C.); liala.baiardi@polimi.it (L.B.); juansebastian.rodriguez@mail.polimi.it (J.S.R.R.)
* Correspondence: marzia.morena@polimi.it; Tel.: +39-02-2399-5189

**Abstract:** The phenomenon of urbanization of cities has been the subject of numerous studies and evaluation protocols proposing to analyze the degree of economic and social sustainability of development projects. Through careful research and synthesis of the theoretical framework regarding residential properties' performance measurement and forecasting, this paper goes deeper into the proposition of property development as an asset class that represents the biggest share of the Italian property market and yet is avoided by the big portfolios. The analysis model was applied to the city of Milan and its Metropolitan Area. The method is based on the development of correlation indices to evaluate different behaviors, through time and a Geographic Information System (GIS) based on the Hedonic Price Method (HPM). Results from a hedonic model estimated for several recent years suggest that, depending on the particular view, the relation between the rent/price performance and the different external and intrinsic variables can represent a useful parameter for evaluating the feasibility of different real estate investments.

**Keywords:** real estate market; residential property; hedonic price method (HPM); urban sustainability; urban transformation





## 1. Introduction

The nature of the property market causes the construction of predictions to be dependent on many factors that are often ignored or undervalued, as well as on the already existing factors that determine less perceptible changes. According to various international research works carried out at an urban level, the true nature of the change in real estate values lies in trends that may be due to the simple perception of the population [1,2], a complex urban process [3,4], or, on occasion, situations where the price system may not immediately reflect the changes suffered by the area in recent times [5,6].

The possibility that residential Real Estate might not be efficiently upraised was given a strong foundation by Case and Shiller (1989) [7]. The authors state that there is a correlation and predictability in the average cost of housing prices but establishes the idea of a possible correlation between future prices with immediate rents as a natural behavior of residential real estate. When considering residential properties, predictability states and reinforces the idea of a sequential correlation [8].

The susceptibility of house prices, and the model with which such prices are calculated, are presented by Poterba (1984) [9] stating that a perfectly informed model of house prices may have positive shocks followed by declining prices and vice versa; this is related to the lags in the supply of new assets. Even if these scenarios are known and incorporated into the model, the nature of the market creates a serial correlation and then, even if these lags are well known and fully incorporated into expectations, "they still create serial correlation and (somewhat) predictable house price movements in reaction to shocks" [10].

Poterba [9] presents an asset market model for residential real estate and he estimates how changes and inflation rates affect the equilibrium of prices. Then, making a model for deducting changes in Real Estate, he includes the effectiveness of the market, like Campbell (1999) [11]; if markets are efficient, value/income proportions ought to be decidedly connected with resulting profit (and value) growth.

Over the past 10 years, extensive research has attempted to evaluate the dynamics involved in metropolitan cities by comparing them. Most of these studies have defined synthetic indicators to measure urban smartness [12] as well as its dimensions and productivity. Kitchin et al. (2015) and Vanolo, (2015) [13,14] have found that although indicators are an effective means of describing complex phenomena and supporting decision-making processes to define adequate urban strategies and actions, sometimes they do not allow measurable elements such as social, demographic, and cultural differences between cities.

The behavior of the real estate market and, in particular, the residential sector, in a large city varies from that of small, medium-sized, and large cities; it is necessary to consider the main differences due to the size of the sector and the dynamics of a metropolitan city. Moreover, it is necessary to add that housing markets are not homogenous across metropolises. Aggregating data at the national level may disguise the true volatility at a local level that homeowners face and care about [15]. In the same way, the areas of a metropolitan city may not reflect the general behavior of the neighborhood or even the sector.

It is precisely this context that the Research Project of Relevant National Interest (PRIN) "Metropolitan cities: territorial economic strategies, financial constraints and circular regeneration" includes their research results (the Projects of Relevant National Interest (PRIN) Are Research Projects Funded by the Italian Ministry of Education, University and Research (MIUR) with the Aim of Strengthening the National Scientific Bases, Also in View of a More Effective Participation in European Initiatives Relating to the Framework Programs).

This research project involved four different research units (UdR): Architecture, Built Environment and Construction Engineering Department of Politecnico di Milano, Iuav University of Venice, University of Bari Aldo Moro and the University of Naples Federico II. The three-year research project aims to investigate the evolution over time of the relationships between the central city and the metropolitan suburbs through census data of the functions and activities hosted. It also intends to deepen the trend of public investments, taxation relating to the metropolitan city, and finally the analysis of the profile of land and real estate income from the center to the periphery of the metropolitan cities.

One of the aims of the research is to demonstrate the trends in ten Italian metropolitan cities and, for Politecnico di Milano research unit, the in-depth study and analysis of the trend of real estate values in the Metropolitan City of Milan.

The analysis of land and real estate trends started in one of the first phases of PRIN Project (Baiardi et al. 2019) [16], within which the variations in price and rental value were analyzed, as well as the capitalization rate.

The research continued with an in-depth analysis of the trends in real estate values in the residential sector and of the factors that can influence them. It is precisely on this theme that this paper focuses.

The research therefore provided the preparation and processing of data through the creation of special indices used to monitor and map residential real estate performance given by data pertinent to the Real Estate Market, making use of data provided by the Agenzia delle Entrate (Agenzia delle Entrate—Revenue Agency—is the Italian Revenue Agency, a non-economic public body that operates to ensure the highest level of tax compliance. It is mainly responsible for collecting tax revenues, providing services and assistance to taxpayers and carrying out assessment and inspections aimed at countering tax evasion. It also provides cadastral and geocartography services, manages all the payment to public administration, handles the e-invoice for all the public authorities and the Health Insurance Card); as well as the identification of socioeconomic factors related

to these trends. Using real observations of housing transactions in the residential market, from 2006 to 2017, the analysis conducted aims to identify the possible future scenarios of the residential property market of the metropolitan city of Milan in specific areas of interest.

## 2. Methodology

The paper aims to identify, through an analysis of price variations, external and intrinsic characteristics, the possible future scenarios of the residential real estate market of the metropolitan city of Milan in specific areas of interest, building-specific tools for decision-making, and analysis of risk.

The construction of a model to analyze trends and investment opportunities in the real estate market of the metropolitan city of Milan involves several steps: The analysis of the previous conditions that have led to the current market situation through the evaluation of the series of existing data (Residential transactions occurred between 2006 and 2017), the assumption of certain parameters and general conditions as well as the analysis of the dynamics existing between the city and its metropolitan area; thus determining the areas of greatest potential to carry out a specific analysis of the expected future changes in these areas, determined by the city's urban plans as well as by investment trends, private projects, and changes in the inhabitants perception of each zone.

Understanding the dynamics of public and private investment in the city was initially evident through the analysis of real estate growth or reduction trends [17] and performance of different factors external and intrinsic to the market, and then proposing an investment scenario [18].

In order to achieve this aim, the research involves the adoption of Hedonic Pricing Method (HPM) proposed by Rosen [19] in the city of Milan in order to quantify the impact that the features (structural, neighborhood, and accessibility characteristics) of a residential property have on its real market price. For this purpose, a set of 352 transactions occurred in the third quarter of 2019 within the urban limits of the metropolitan city of Milan have been used. GIS (Geographic Information System) has been used to analyze the location properties on the map and to measure distances between each property and the variables described in Section 6. Subsequently, the model will be used for evaluating the impact on prices and rents (the model will be run with both) that selected private and public urban projects will have in the behavior of the city. In this scenario, once the initial model is built, the current situation will be modified with a second scenario and prices and rents will be recalculated according with the results of the first model, and these results will be used in the investment scenarios.

Through a statistical regression in cases where a total or partial mapping of the information is possible, a model was constructed (Figure 1).

In addition to the hedonic model, an important starting point for the completion of the analysis was to analyze not only prices but also rent, thanks to the document presented by Lee, Seslen, and Wheaton [20], where a direct relationship between rent and prices is established throughout subsequent prices.

In the proposition of this analysis, the main idea is that prices and rents are the correct comparison for a test of autocorrelation. The rent is considered as a linear combination of its various attributes (hedonic equation). Subsequent price appreciations will be considered in a specific area, in a panel of at least 5 years.

The first endeavor to use value/rent proportions for testing housing market efficiency was developed by Meese and Wallace [21]. These authors examine the ratio of average between house prices and average rents and find some positive correlation to subsequent price appreciations.

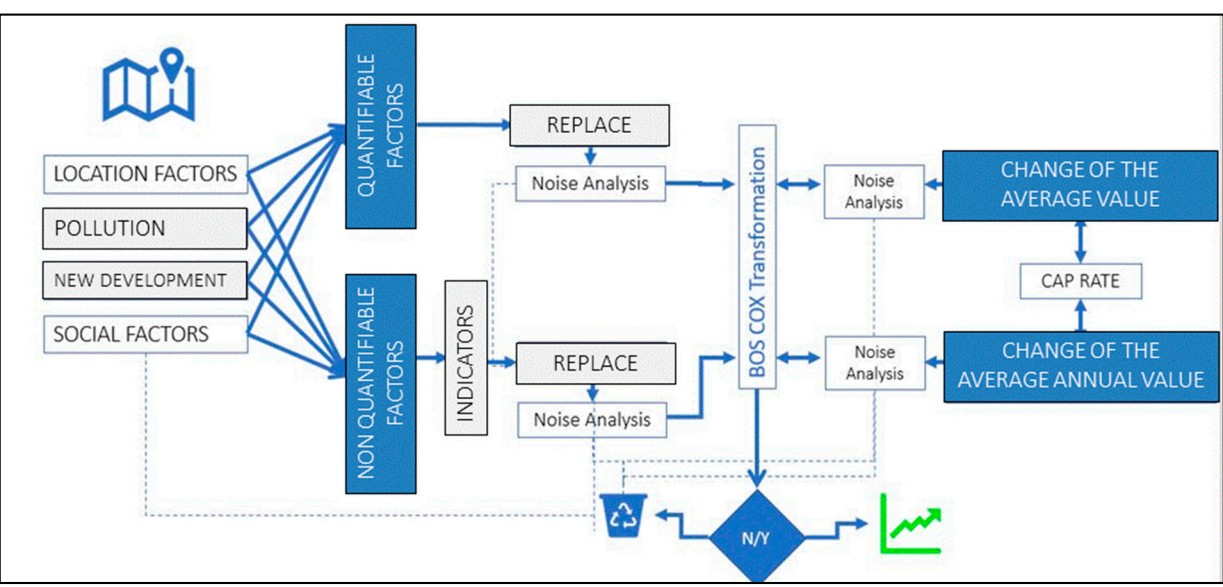

**Figure 1.** Model algorithm.

A study by Campbell et al. [11] attempts an exhaustive time examination of value/rent rates and presumes that there is solid proof of a time-relation in metropolitan cities. At the point when rents are high, they will in general fall, with most of the alteration being made by prices instead of rents.

If rents change over time, the volatility of prices should be less than that of rents since they represent the discounted value of expected future rents. Rents play a more important role when adjusting actual conditions efficient price rent ratios are the highest when oscillations have driven rents to a (temporary) bottom, and lowest when rents are at a (temporary) peak. According to Abel, Andrew and Blanchard, Olivier [22] "Hence regardless of whether the rents in question are smoothly trending up or down, or oscillating in some manner, efficient price/rent ratios are positively correlated with near-term subsequent rental movements. When prices are the present value of rents the correlation will extend to prices as well".

According to Lee, Seslen, and Wheaton [20] a unit's predicted price using its current characteristics in a hedonic model has a rent as input. Hence, the residuals from this hedonic equation should reflect the omitted determinants of rent and the expected future growth. This way of modelling will test if housing Hedonic equation residuals have a significant positive correlation with subsequent price growth.

The simulations are raised specifically for the areas of interest and are applied to the models built to identify possible changes in real estate or rental values. For such scenarios a risk study was developed, considering the accuracy of the models and characteristics of the area, the feasibility of investments analyzed depending on capital expenditures, and the profiles of possible investment funds or individual investors.

The model permits us to:

- Identify and interpret the general trends of the residential property market and identify the areas that are most interesting for an in-depth analysis of trends through a self-constructed forecasting model.
- Identify and quantify the impact of the different socio-economic and spatial variables on the price treated in the areas of interest, under the critical issues and constraints.
- By simulating future scenarios in the real estate forecasting model, pose the different future situations that the residential market could present, as well as develop a risk analysis within the different areas of interest.
- Identify in which areas price/rent ratios are positively correlated with subsequent rent (and hence price) growth, raising future investment scenarios.

- Identify the different possible responses that the residential property market may suffer in the future due to general changes in the city, its socio-economic conditions, and moreover under the effect of public and private projects, in order to build a series of investment scenarios.

Through the use of GIS tools and regression models it was possible to identify areas of interest and general trends, in addition to identifying the main challenges for the elaboration of a more detailed study that would allow future analysis.

The paper is partitioned into following sections:

- Real estate forecasting will analyze different methods for predicting prices and trends, defining its limitations and advantages depending on the availability of data. Moreover, a method for deducting general behaviors from price/earnings ratios and its positive correlation with subsequent earnings (and price) growth will be introduced.
- Data gathering and description summarize the sources and treatments that were given to data, the entities in charge of providing real estate official information, and the sources of different crucial variables that were considered in the model.
- Econometric modeling of the residential properties in the metropolitan city of Milan is a statistical assessment of the real estate profile in the Metropolitan City of Milan, only focused on the micro areas that were already identified in the PRIN "Metropolitan cities: territorial economic strategies, financial constraints and circular regeneration" project, analyzing also the current situation of residential properties in Italy and Milan. This section defines the parameters used in the creation of a model able to be modified with further assumptions for future investment scenarios that will be postulated.
- Analysis and interpretation of the results in order to evaluate future scenarios and investment opportunities in residential properties in the metropolitan City of Milan. This last section completes the paper with investment scenarios using the analysis framework for the real estate profile. The different scenarios consider public investment in new infrastructure, the impact of new urban developments in the city, and also the effect of private projects that strongly affect the current state of specific areas in the city raising different opportunities for investing in specific areas. This effect as is seen might already be happening since the effect price/rent ratio reflects a faster response of the market than just analyzing the timeline of acquisition prices. This section provides evidence of investment properties with indications about the advantages and risks of the transactions.

## 3. The Metropolitan City of Milan (MCM) as a Privileged Study Context

The importance of geographically defining the study to be performed is based in the light of overall regional economic conditions. In this sense, the analysis should "provide background on the location of the site within the metropolitan area, for example, the distance to downtown, to the airport, and to other regional draws" [23].

In order to analyze the Metropolitan City residential market, it is important to understand the scale of the analysis and the specificity of the information available [24] as well as the scale of the objectives of this research.

The Metropolitan City of Milan (MCM) was established on 8 April 2014 lying inside the Province of Milan as a result of the entry into force of Law 56/2014 (art. 12) [25]. According to its metropolitan city charter [26], the MCM aims to express the best of the culture of government and the administrative experience of the municipalities of its territory, each carrying stories and traditions in an integrated and polycentric framework that respects their identity and enhances their participation. The MCM wants to make administrative simplification its working method. The role of the new body and our common political and civil commitment are defined around these challenges.

The fundamental functions of the MCM are established by Section 85 of the Law n. 56 of 7 April 2014 [25], in force since 8 April 2014, and by Regional Law 92/2015 [27], amended into Law 32/2015, published in the Regional Official Bulletin on 16 October 2015. In particular, Law no. 56 of 2014 [25] (in articles 85 and 86) underlines that among

the fundamental actions exercised by metropolitan cities, there are: Provincial territorial planning for coordination, as well as protection and enhancement of the environment, for the aspects of competence (letter a) of art. 85); and the care of the strategic development of the territory and management of services in associated form based on the specific characteristics of the territory itself (letter a) of art. 86).

The former Metropolitan City of Milan (MCM) is made up of 134 Municipalities, covering an area of 6827 square kilometers and has 3,196,825 inhabitants, almost a third of the regional population. There are 1,337,155 inhabitants in the municipality of Milan (over 40 percent of the former provincial population) and the territory is classified as entirely flat; with 1500 kmq of surface, the MCM is the second urban area in Italy [28].

The MCM is divided into different homogeneous areas with similar socioeconomical characteristics and equivalent to areas with similar rent and sell value characteristics [26].

Given that the behavior of the residential sector in a large city varies in a different way from that of a small, medium-sized, and large cities, it is necessary to consider the main differences due to the size of the sector and the dynamics of a metropolitan city. Besides, housing markets are not homogenous across metropolises [29].

It is critical to have a full model when considering costs and value changes across jurisdictional submarkets in a metropolitan area that is subdivided in homogeneous areas like Milan. A standard model of metropolitan housing submarkets accepts that a pre-determined number of households look for a market over a set number of areas. Every area has a fixed amount of supply. Family units get utility from the market, an amenity associated with being located in a determined area, in other words, each homogeneous area has special characteristics that make it attractive to house buyers [30].

The flexibility of the metropolitan market is an important issue to be considered, since offer and demand may vary asymmetrically. Moreover, for a product as diverse as real estate, in metropolitan cities a considerable reduction in population and the relative low elasticities of supply and demand might be expected. "*It is common in metropolitan areas that asymmetric price reactions growth in population numbers has no significant effect on price*", whereas declining population significantly lowered prices, hence "*rental and the property sectors are characterized by low price elasticity of demand, indicating that significant real estate price decreases can result*" [31].

The dynamics in metropolitan urban communities in western cities as Milan are not the same as in developing nations, since critical beneficial outcomes are shown for disposable income and construction costs, and the reaction to the creation of supply is asymmetrical, meaning that population growth and the resulting increases in demand have no significant effect on price. However, a declining population leads to significantly reduced prices [25].

Several factors influence the attractiveness of one real estate asset compared to others in the same metropolitan market. Some analysts start with an overview of the demographic indicators of the metropolitan area or add indicators that describe the conditions of the local market, thus facilitating the comparison and the regional or metropolitan contrast with the conditions of a specific submarket, or the area of interest for a project. "*At a minimum, demographic and economic data should go back as far as the preceding decennial census. However, once the most recent census is more than a few years in the past, the analyst will need to provide more current estimates and projections*". This is the main reason why this analysis also considers transactions made after the main database available (OMI transactions until 2017) needs to be updated to the latest available [23].

A metropolitan diagram must incorporate data on the construction sector. For housing, real estate studies, building permits, and construction volumes have to be tracked, ideally with isolated arrangements for single-family and multifamily projects. The economic conditions of each neighborhood may not accurately reflect national patterns: Not every metropolitan homogeneous zone profits by a national financial blast. Accordingly, Real estate market studies give more importance to regional and metropolitan economic indicators than to nationwide statistics [23].

## 4. Data Gathering and Description

Even if residentially focused, the Italian market has no records for specific subsectors or neighborhoods of a city delivered by institutional bodies. Analyzing who produces Real Estate Information in Italy, governmental entities in charge of compiling the information, trade associations, and consulting companies are the most interesting data providers. The main problem being the low level of connection among all these bodies, there is no systematic trade of information. Among the official entities, there is the Osservatorio del Mercato Immobiliare (OMI) an institution that is reliant on the Agenzia delle Entrate.

The OMI is responsible for collecting and processing technical and economic information relating to property values, the rental market and annuity rates, and the publication of studies and calculations for the statistical enhancement of the Agenzia delle Entrate's archives [32].

In the OMI's dataset structure, in order to homogenize the data, the metropolitan city of Milan (and the city of Milan) has been divided into different areas (OMI zones), which are defined as a continuous portion of the municipal territory that reflects a homogeneous sector of the local property market, in which there is uniformity of appreciation for economic and socio-environmental conditions [33].

The area is defined as a homogeneous segment of the local real estate market, in which there is uniformity of appreciation for economic and socio-environmental conditions. It is formed on the basis of the deviation, between the minimum value and the maximum value, not exceeding 50%, found for the prevailing typology, within the residential destination.

After the 2014 update, the municipality of Milan was divided into 41 reference real estate areas, the so-called "OMI Zones", 14 less than in 2006 (The territorial areas of the OMI areas are subject to a ten-year review process, in order to verify their consistency with the urban development of the territory and with the rules for the formation of municipal zoning. The updating of the articulation of the municipal territory by homogeneous areas must be justified exclusively in structural changes of the territory and / or in a changed reality of the local market, found through appropriate and detailed territorial analysis, following the criteria and operating methods set out in the Manual.The updating of the municipal zoning is released every semester with the validation of the cartographic archives according to the following terms and can take place in the following ways: Partial update; 10-year update.The updating of the perimeters of the OMI areas must also be ensured in the event of changes in the cadastral map and / or in the presence of errors. In such cases, the following procedures are applied: Realignment; grinding) (Figure 2).

In 2014, a general review was carried out of the territorial areas (OMI areas) within which the prices of the properties are defined. This operation, which ended in the second half of 2014, was necessary to incorporate the changes in the urban and economic fabric.

On that occasion, improvements were made in terms of a uniform methodological approach, internal controls, and fine-tuning in the boundary delimitations, starting from the second half of 2014.

For this reason, the comparison between the quotations for the second half of 2014 and those of the previous semesters for individual areas is not always possible, where the perimeters of the territorial areas of reference have changed substantially.

The change of the homogeneous divisions is evident in the figure below. Given the considerable change in the area and their shape and coverage, the average prices from 2006 to 2013 will not be considered for the analysis in this dissertation. A study of the consequences of the change of OMI areas was already presented in the context of Research Project of Relevant National Interest—PRIN "*Metropolitan cities: territorial economic strategies, financial constraints and circular regeneration*" [34].

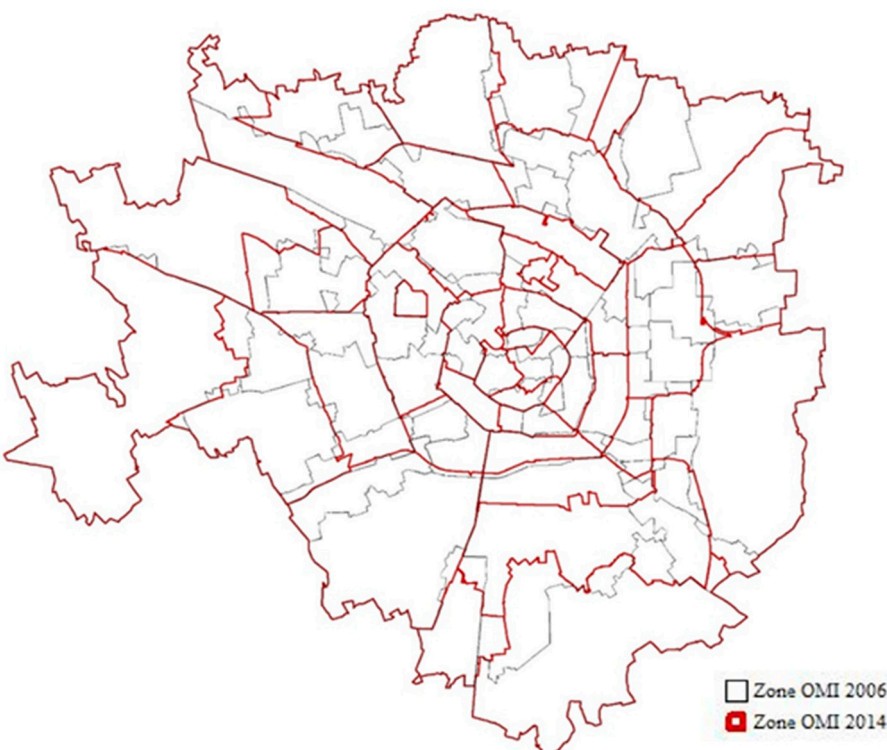

**Figure 2.** Difference between Osservatorio del Mercato Immobiliare (OMI) areas in 2006 and 2014.

The real estate value in the municipality of Milan is given by the type, age, and architectural characteristics that distinguish the single area of the municipality of Milan.

In order to build the econometric model, it was important to find the relationship between the variation of socio-economic factors with the change in the real estate value in the residential category of the entire Metropolitan City of Milan with an analysis period of 11 years, from 2006 to 2017, evaluating the importance of the various socio-economic characteristics as drivers of the change in the real estate value.

This analysis attempted to evaluate the change from year to year in the average sales value of residential real estate units, which is why a standardized transaction database, provided by the Agenzia delle Entrate, is used.

As presented in the paper "Real estate market values and land revenue analysis in the Metropolitan city of Milan" [16], the division of the metropolitan city into homogeneous zones shows a high heterogeneity of characteristics between the 2006–2013 classification and that of 2014–2017. This phenomenon is present for all groups. The analysis must be concentrated only on the variation in the periods before and after the update, and the data for the years 2018 and the first half of 2019 will also be taken into consideration by the revenue agency for greater accuracy. In this sense, the new analysis and model was built only with the most recent data, and no data before 2014 were considered.

As for the analysis on the influence of some geographical, physical, and socio-economic factors, the state of conservation of the building is undoubtedly the factor with the most significant impact, followed by the quality of the neighborhood, and the distance to the center is confirmed as an important exchange rate factor for listing; the proximity to the city's large parks and the metro have an important positive effect on prices. The change in status of the neighborhood, along with green areas and accessibility, are the geographical factors identified that could mean a change in prices, given that the other spatial characteristics are fixed. Other exchange factors must be seen individually for each district (e.g., large real estate developments, market dynamics).

However, following this analysis, the limitations given by the lack of punctual reference for transactions were evident. Although the database of the Real Estate Market

Observatory aims to provide guidelines for all operators in the sector, it is necessary to study geolocated data in a point to better study the phenomenon.

Within the project, the division of the city into neighborhoods was considered, given that the databases and sources of qualitative data are based on this division to report results.

The municipality of Milan is divided into 88 districts, not directly connected to the delimitation of homogeneous areas except in some cases (e.g., Tre Torri). For this reason, it was necessary to take into consideration the division into city districts; the same methodology used to compare the OMI areas before and after the update was therefore applied to the districts (Figure 3):

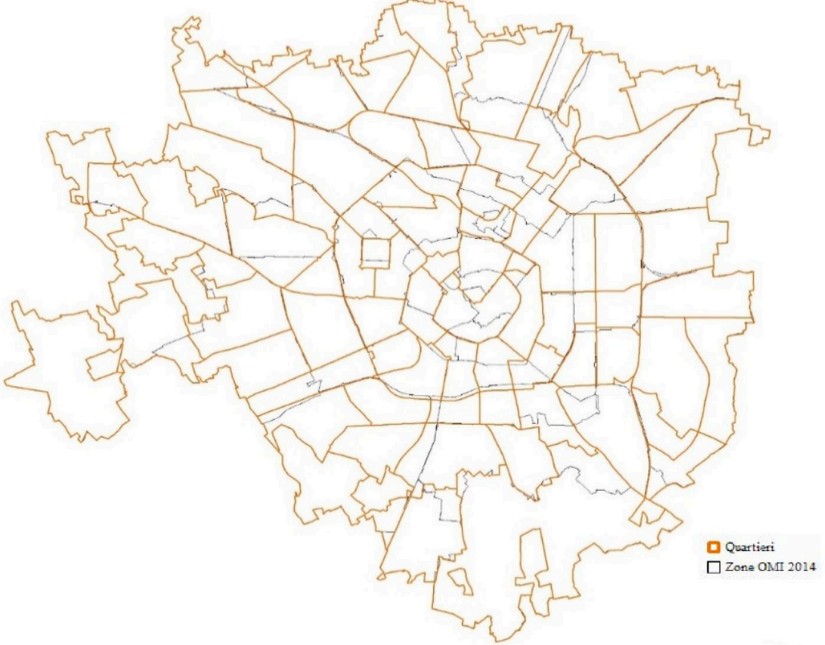

**Figure 3.** Discretization of real estate values of Zona OMI 2017 in the 88 districts of Milan.

Although there are a wide range of techniques for managing the way towards building a real estate model, the following methodology was implemented, as performed in the Hedonic study by Vries et.al. (2009) [35]:

1. Economic Real Estate theory.
2. Data gathering.
3. Formulation of the econometric model.
4. Modeling prove and noise data elimination.
5. Representativeness evaluation.
6. Reformulate model/Interpret model
7. Use model for investment proposals.

## 5. Econometric Modeling of the Residential Properties in the Metropolitan City of Milan

For confronting statistical deficiencies recently experienced and connected with transaction-based indices, two methods have been developed: The Hedonic Price Methodology and the repeat-sales regression methodology [36]. The hedonic pricing method depends on the principle that a property is a complex with plenty of sections which, regardless of whether or not they can be traded independently, have their very own valuable elements. Computing the value of all attributes for every trade, everything being equal—both physical (size of the land, size of the building, development period, number of rooms, construction quality, and so on) and localization (area, quality of the area, accessibility, noise, and so forth)—it is conceivable to generate an index by researching the value for a given property and by regression to gauge each characteristic [37].

Fundamentally, the process depends on the representation of the market through econometric models. The reliant variable is the cost of the property or the relative price of utilization at various occasions of time. The simplest model is to consider the state of the buildings, their location, and so on.

We can compose y = f(X) + ε, indicating with y the value per sqm of the real estate units, with X the vector of the specific attributes observed and with ε the stochastic errors of the model with the theory that they are regularly distributed with consistent variance, ε~N (0, σ2).

The calculated parameters, achieved as inherent costs ascribed by the market to individual characteristics, can be applied to a lot of standard attributes to acquire a value that gives a gauge of the cost that would have been seen without fluctuation in the features of the properties. The valuation of a specific housing unit can be acquired by applying the intrinsic costs to the attributes of the residence.

The utilization of this methodology allows one to develop indices of steady quality dependent on transaction costs, with, however, the utilization of a regression model that permits the attributes and consequently the heterogeneity of real estate to be considered. From an econometric perspective, the building quality is studied and thus the value variety of a property whose qualities would be consistent after some time is featured.

There are two fundamental techniques for the construction of a real estate index dependent on the hedonic method:

(a)    Regression model will be determined for every period;
(b)    Single regression model utilized for all periods concerned.

The principal strategy includes performing a hedonic regression for every period by following the development by time of the estimation of a common property (or a portfolio of properties) and evaluating the value in every period, utilizing the coefficients of hedonic regressions.

The principle limitation of the hedonic technique is connected to the complexity of the econometric models that must be applied to guarantee the nature of an index. What is more, specific consideration ought to be paid to the utilization of these techniques: Specifically, it ought to be guaranteed that all the most significant factors have been incorporated into the model, yet additionally the model's illustrative factors are not very interrelated and that the estimations of the diverse variables are estimated precisely. The subsequent restriction concerns the quantity of the databases necessary for the advancement of such indices: The market inclusion ought to be as significant as would be prudent. It is subsequently important to persuade the holders of this data (specifically public accountants, mortgage banks and real estate experts) that it is helpful to make it accessible so as to know value increases on the residential markets.

The general objective of the model built by this dissertation is to interpret residential real estate prices as a variable dependent of several characteristics, the theoretical model follows the alignments of a Hedonic Price Model (HPM) and several variables will be defined, indexes for residential real estate performance will be used, as well as a new indicator for reflecting the independent variables. The information will be collected from official institutions and other sources that gather other types of information. The estimation indexes were built in a such a way that they could reflect the current situation of every location in the most suitable way. The statistical evaluation of the model considers the assumptions made in it and describes data properly, that will be then interpreted and analyzed.

The model required for the most significant part of the analysis includes various distinctive general arrangements of tasks:

(a)    An assortment of record organizations and information sources.
(b)    Normalizing data and avoiding inclusion of non-significant information in the model.
(c)    Transformation applying scientific and factual tasks to gatherings of datasets to infer new datasets (e.g., amassing a huge table by bunch factors).

(d)   Modeling and calculation. Associating the information to the model, AI calculations, and other computational tools as GIS, used several times in this dissertation.

(e)   Presentation. Making intuitive graphical representations of the results.

This project performs an HPM in the city of Milan in order to quantify the impact that the features (structural, neighborhood, and accessibility characteristics) of a residential property have on its real market price. For this purpose, a set of transactions within the urban limits of the metropolitan city of Milan has been used. GIS has been used to analyze the location properties on the map and to measure distances and areas of the neighborhood characteristics of the transactions. Subsequently, the model will be used for evaluating the impact on prices and rents (the model will be run with rents and costs) that selected private and public urban projects will have in the behavior of the city. In this scene, once the initial model is built, the current situation will be modified with a second scenario and prices and rents will be recalculated according with the results of the first model. These results will be used in the investment scenarios.

Through a Box Cox statistical regression in cases where a total or partial mapping of the information is possible, a model was constructed.

In addition to the hedonic model, rent has also been analyzed, referring to what is presented by Lee et al. (2015), where a direct relationship between rent and prices is established throughout subsequent prices. The main idea of authors is that prices and rents are the correct comparison for a test of autocorrelation. The rent is considered as a linear combination of its various attributes (hedonic equation). Subsequent price appreciations will be considered in a specific area, in a panel of at least five years.

The first endeavor to utilize value/rent proportions for the testing of housing market efficiency was developed by Meese and Wallace (1995) [21]. These authors examine the ratio of average between house prices and average rents and do find some positive correlation to subsequent price appreciations.

If rents change over time, the volatility of prices should be less than the one of rents since they represent the discounted value of expected future rents. Rents play a more important role when adjusting actual conditions, efficient price rent ratios are the highest when oscillations have driven rents to a (temporary) bottom, and lowest when rents are at a (temporary) peak. "Hence regardless of whether the rents in question are smoothly trending up or down, or oscillating in some manner, efficient price/rent ratios are positively correlated with near-term subsequent rental movements. When prices are the present value of rents the correlation will extend to prices as well" [38].

According to Lee et. al. (2015) [20] a unit's predicted price using its current characteristics in a hedonic model has rent as input. Hence, the residuals from this hedonic equation should reflect the omitted determinants of rent and the expected future growth. This way of modelling will test if housing Hedonic equation residuals have a significant positive correlation with subsequent price growth.

The methodology worked with three categories of independent variables: Structural characteristics of the house, neighborhood environmental characteristics, and accessibility characteristics, following the categorization of Fanhua et.al. [39,40].

The data collection process consisted of private housing transactions in the city of Milan, occurred in the third quarter of 2019, taking into account the price per square meter of the properties, their condition, the internal characteristics of the property, and its location, together with the capitalization values obtained from the OMI analysis, in addition to the average yearly increase of a stable area x (Table 1).

For the calculation of an initial cash flow, the following parameters must be considered:

- Taxes correspondent for a secondary house: 10.156% of the gross annual rent.
- An average of the market for administration fee of about 0.8% of the gross annual rent.
- Indexation at 100% of ISTAT values, taken from the Focus economics magazine
- Sale cost 3% of the final price.

**Table 1.** Definition of the average yearly (example).

| Period | Indexation | | | | | | | | | | |
| | Year 1 | Year 2 | Year 3 | Year 4 | Year 5 | Year 6 | Year 7 | Year 8 | Year 9 | Year 10 | Year 11 |
|---|---|---|---|---|---|---|---|---|---|---|---|
| Value | 0.78% | 1.02% | 1.23% | 1.41% | 1.58% | 2.00% | 2.00% | 2.00% | 2.00% | 2.00% | 2.00% |

A Geographical Information System (GIS) has been used to analyze the location properties on the map [41] and to measure distances and areas of the neighborhood characteristics of the transactions.

Several variables were introduced into the model: Demographic, economic, spatial, structural (intended as intrinsic to the property), considerations of taxation, and other variables [37].

The demographic variables are defined based on data provided by ISTAT (2017) [28] (Table 2). They are represented in the following table where the correlation of these indicators has been defined and studied for each neighborhood. For changing quotations from neighborhood to areas and vice versa, GIS software was used.

**Table 2.** Demographical variables considered (Source: ISTAT, 2017).

| ID | Scope | DEN | Indicator | Definition |
|---|---|---|---|---|
| 1 | Territorial | D | Housing density-Ab/Km$^2$ | Ratio between the population resident in the area and the relative area in Km$^2$ |
| 2 | Mobility | Ic | Centrality index | Ratio between the number of commuter flows entering the area (net of commuters residing and working in the area) and the number of commuter flows leaving the area (net of the same amount). |
| 3 | Demographic | Iv | Age index | Percentage ratio between the resident population aged 65 and over and the population in the class 0–14 |
| 4 | Demographic | Is | Incidence of foreign residents | Ratio between the foreign population and the resident population per thousand |
| 5 | Education | Insm | Index of non-completion of the first-cycle secondary school | Percentage ratio between the population in the age group 15–52 who did not graduate from the lower secondary school and the total population of the same age group |
| 6 | Economic | Td | Unemployment rate | Percentage ratio between jobseekers and the labor force |
| 7 | Social vulnerability | Neet | Incidence of young people outside the labor market and training | Percentage ratio of the resident population in the 15–29 age group in non-professional condition other than a student and the resident population in the same age group-enlarged NEETs |
| 8 | Social vulnerability | Fde | Incidence of families with potential economic risk | Percentage ratio of the number of families with children whose reference person has until |

Two variables defining the economic situation of the householder and the area were introduced into the mode: Average income and the commercial vocation of the area.

Average income of the residents: The model first analyzes the relationship between price and rents with the average income of residents. After the analysis of income and rent, even the change in income declared IRPEF does not seem to have a general direct correlation for all the municipalities of the Metropolitan City of Milan with the change in the residential real estate value; it can be seen how the territorial generalization cannot be

applied in this context. As can be seen in the graph (Figure 4) where every municipality integrating the Metropolitan City of Milan was represented as a point considering its IRPEF and its average housing price, there is no direct relation between the IRPEF and the average price of residences.

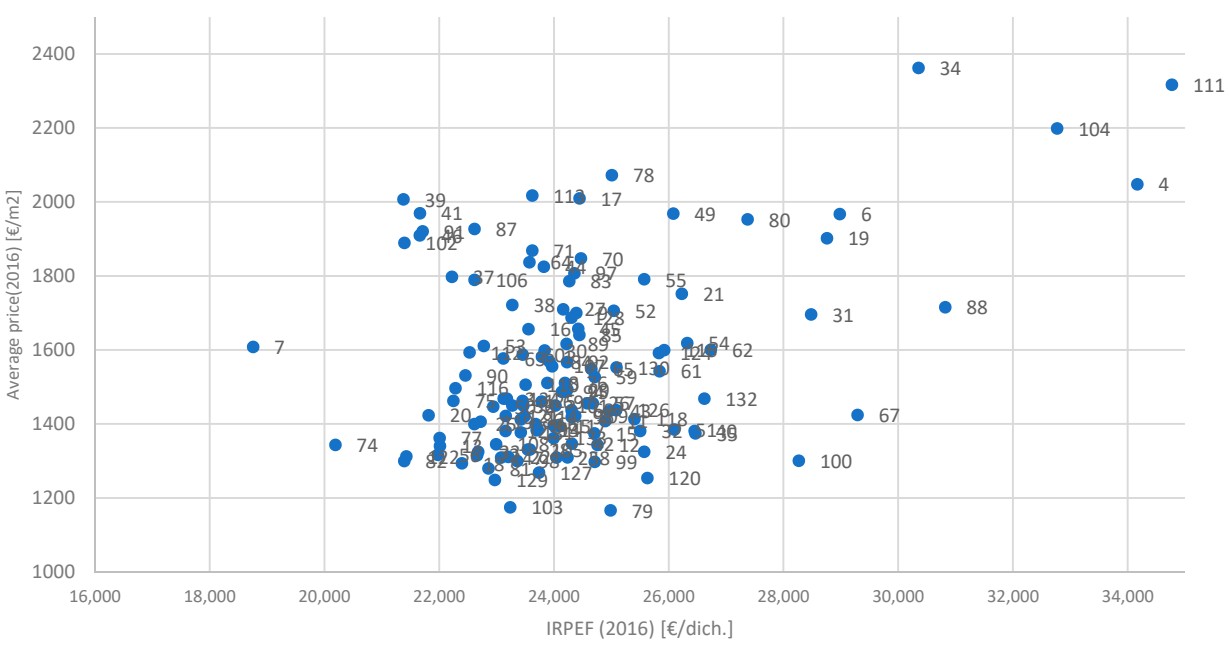

**Figure 4.** IRPEF (Imposta sul Reddito delle Persone Fisiche) vs. OMI average price, municipalities of the Metropolitan city.

Commercial aim: This variable represents the importance that is given to areas that are active for shopping, stores, restaurants, etc., and is measured by the presence of the "Districts of Commerce", as defined in the geoportal of the city of Milan.

As regards the spatial variables, the research adopted the same variables adopted by the IMO. Therefore: condition; distance to the center; average year of construction; average number of floors; presence of commercial services; presence of green spaces; presence of public transportation; parking availability; road connections; commercial vocation; and finally, quality of the area.

Therefore, it is important to understand in which magnitude each of these characteristics affect the price of the area: Hence the single transaction, the characteristics that refer to the intrinsic state of each individual property (prevailing state, average year of construction, average area, and average number of floors) are already considered in the "condition" of the building. The other characteristics depend on the spatial arrangement of each transaction, to identify which factors are more important a qualitative analysis is proposed for the city of Milan.

This analysis is conducted with the aim of identifying the factors that, in addition to the special characteristics of each district (new developments and real estate projects), can influence the price change in some way.

According to Istat (2017) the quotations of neighborhoods, zones, and municipalities are closely correlated with their distance from the city center.

Presence of public services is measured through the presence of public healthcare infrastructures, schools, and universities, as reported in the geoportal of the municipality of Milan.

Presence of public green space is measured through different indicators: Parks Proximity Index: Influence of each green area classified as Urban Park (ParkP) over the distance to each transaction.

Urban vegetation proximity index (GrVegP): Influence of the areas classified as urban vegetation (ISTAT4) over the distance of each transaction.

Special Green areas proximity index (SGrAP): Influence of the areas classified as school gardens, vegetable gardens, cemeteries, and public gardens (ISTAT 5) on the distance of each transaction.

Urban Green area index: Influence of the areas classified as equipped green areas and historic green areas (ISTAT 1,3) on the distance of each transaction.

Regarding the level of transport services, this is measured as Metro Proximity Index (Metro PI): Indicator of the proximity of metro stations.

Status of the area: Weights from 1 to 6 (the highest, plus status) classify the different areas of Milan, based on their status considerations, in function of urban land use.

Iconic Places Proximity Index (IconPI): Proximity to iconic attractions for tourists in the city, with a weight of 5 to 1 based on the number of their annual visitors.

Regarding the intrinsic property variables or structural characteristics, as said before, one limitation of the model is the fact that several characteristics of each house are in every OMI area are reduced to one single state where buildings were given a number for their corresponding condition (state); 4 for new, 3 for Perfect (renovated), 2 for Good, 1 for Bad (to be renovated).

The data administered by the Agenzia delle Entrate is measured according to their 'status': Poor, normal, and excellent. The listing is then associated with this status. This is an important factor, since it defines the only characteristic that refers to the condition of the apartment. As can be seen in the two maps below, quotations change drastically due to this factor.

Finally, regarding other variables research considered: Contaminated areas Index (ContAI): Influence of each site with soil and/or groundwater contamination (caused by accidental events, spills, and illegal activity. Potentially Contaminated Areas Index (PContAI): Influence of each potentially contaminated site (defined as a site where one or more concentration values of the polluting substances detected in the environmental matrices are higher than the concentration threshold values of contamination) over the distance to each transaction.

Furthermore, Air Quality Indexes: According to the 2018 air quality measuring stations, a density map was built through GIS, assigning a value of pollution for different contaminant agents: $C_6H_6$ (Benzene, a monocyclic aromatic hydrocarbon), CO_8H (Carbon monoxide (CO), indicates the average concentration over 8 h), $NO_2$ (Nitrogen dioxide), PM10 (Powders with diameter less than 10 μm). Values were measured in $g/m^3$.

## 6. Model's Goodness of Fit: Results Analysis

All the collected information from official sources and from the definition of the variables mentioned was modelled through a hedonic model. The sample for the dependent variable, namely the price per square meter of a residential property in Milan, consisted of 352 residential transactions that occurred in the city during the third quarter of 2019 as shown in Figure 5, all properties distributed across the city to give the highest possible coverage, and discretizing all the information into a GIS supported platform.

The discretization of the intrinsic characteristics of each property consisted in the definition of a scale to describe the state of conservation of the property: The house State (State), from 1 to 5, 1 being a decadent state and 5 as an optimal state, the spatial distribution of the property was evidenced through the number of Rooms (Rooms), the year of construction of the building was discretized as Old coefficient (OldCo), the lower the value the older the building, the Distance to the city Center (DistCBD), Distance to a Metro Station (MetroPi), and finally the socio-economic characteristics of the neighborhood were described through the variable Status of neighborhood (NeighSt).

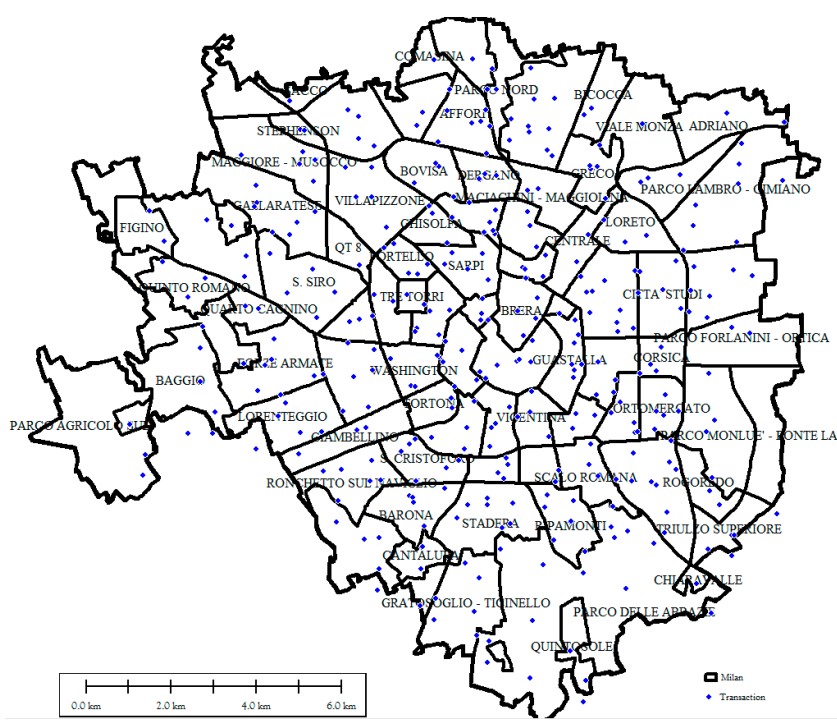

**Figure 5.** Transactions used for the construction of the model.

The model was built by a natural log transformation of the housing price (Y) and a total of 25 variables (X). The first run with all variables can be seen below. Since a non-linear nature of the variables is expected, a logarithmic (log-log) regression was tested (Table 3); log-log transformation implies that coefficients are a percentage change of the variable Y to a percentage change of the variable X.

**Table 3.** First run regression with all the variables.

| Variable | Coeff | Coeff. EE | *T* Value | *P* Value | FIV |
|---|---|---|---|---|---|
| Constante | 10.836 | 0.965 | 11.23 | 0.000 | |
| Bathrooms | 0.120 | 0.114 | 1.05 | 0.299 | 3.59 |
| Rooms | −0.2385 | 0.0956 | −2.50 | 0.016 | 2.84 |
| Old Coefficient | 0.2321 | 0.0837 | 2.77 | 0.008 | 2.24 |
| State | 0.2811 | 0.0948 | 2.96 | 0.005 | 1.39 |
| Status of the neighbourhood | 0.1295 | 0.0664 | 1.95 | 0.057 | 2.07 |
| Distance to the center | −0.369 | 0.108 | −3.41 | 0.001 | 7.19 |
| Degradation Proximity Coefficient | −0.0734 | 0.0204 | −3.59 | 0.001 | 2.36 |
| Big Park Proximity Index | 0.1089 | 0.0462 | 2.35 | 0.023 | 2.05 |
| UrbanVegetation Proximity Index | −0.0175 | 0.0121 | −1.44 | 0.157 | 1.36 |
| Special green areas Index | 0.0587 | 0.0301 | 1.95 | 0.057 | 2.64 |
| Equiped Green Areas Proximity Index | −0.0355 | 0.0272 | −1.30 | 0.199 | 1.96 |
| Contaminated areas Index | −0.0056 | 0.0193 | −0.29 | 0.772 | 1.96 |
| Potentially Contaminated Areas Index | 0.0141 | 0.0218 | 0.65 | 0.520 | 2.58 |
| Metro Proximity Index | 0.0587 | 0.0223 | 2.63 | 0.011 | 2.97 |
| Bus and Tram Proximity | −0.0203 | 0.0296 | −0.69 | 0.496 | 1.74 |
| Train Proximity Index | 0.0398 | 0.0204 | 1.96 | 0.056 | 1.77 |
| Iconic places Proximity Index | 0.0579 | 0.0347 | 1.67 | 0.102 | 10.93 |
| Kindergarden Proximity Index | −0.0403 | 0.0685 | −0.59 | 0.559 | 5.63 |

**Table 3.** *Cont.*

| Variable | Coeff | Coeff. EE | *T* Value | *P* Value | FIV |
|---|---|---|---|---|---|
| Primary schools Proximity Index | −0.1786 | 0.0862 | −2.07 | 0.044 | 7.61 |
| Secondary schools Proximity Index | 0.0446 | 0.0422 | 1.06 | 0.296 | 5.11 |
| Universities Proximity Index | 0.0196 | 0.0248 | 0.79 | 0.434 | 4.48 |
| $C_6H_6$ | −0.0110 | 0.0228 | −0.48 | 0.632 | 1.21 |
| CO_8H | 0.0003 | 0.0149 | 0.02 | 0.985 | 1.45 |
| $NO_2$ | −0.0148 | 0.0320 | −0.46 | 0.647 | 2.44 |
| PM10 | −0.0001 | 0.0137 | −0.01 | 0.993 | 1.38 |

where: Coeff: Percentage change of the variable Y to a percentage change of the specific variable. Coeff. EE standard error of the coefficient. *T* Value: t-statistics value (ratio of the departure of the estimated value of a parameter from its hypothesized value to its standard error). *P* Statistical significance value. FIV: VIF, Variance Inflation Factor.

The first run of the regression presented an R2 adjusted value of 0.723 indicating a 'good fit', yet according to statistical significance rules, the variables with *p* value $\leq 0.05$ prove strong presumption against the null hypothesis. Therefore, using a stepwise forward selection, the following variables were found to be statistically significant (Table 4):

**Table 4.** Second run regression.

| Variable | Coeff | Coeff. EE | *T* Value | *P* Value | FIV |
|---|---|---|---|---|---|
| Constante | 10.860 | 0.758 | 14.33 | 0.000 | |
| Rooms | −0.1346 | 0.0645 | −2.09 | 0.041 | 1.35 |
| Old Coefficient | 0.2006 | 0.0675 | 2.97 | 0.004 | 1.52 |
| State | 0.3811 | 0.0825 | 4.62 | 0.000 | 1.10 |
| Status of the neighbourhood | 0.1446 | 0.0606 | 2.39 | 0.020 | 1.80 |
| Distance to the center | −0.4057 | 0.0787 | −5.15 | 0.000 | 3.97 |
| Degradation Proximity Coefficient | −0.0639 | 0.0158 | −4.05 | 0.000 | 1.47 |
| Big Park Proximity Index | 0.0973 | 0.0373 | 2.61 | 0.011 | 1.39 |
| Metro Proximity Index | 0.0439 | 0.0194 | 2.26 | 0.028 | 2.35 |
| Train Proximity Index | 0.0352 | 0.0178 | 1.98 | 0.052 | 1.41 |
| Iconic places Proximity Index | 0.0683 | 0.0254 | 2.69 | 0.009 | 6.08 |
| Primary schools Proximity Index | −0.1080 | 0.0443 | −2.44 | 0.018 | 2.09 |

where: Coeff: Percentage change of the variable Y to a percentage change of the specific variable. Coeff. EE standard error of the coefficient. *T* Value: t-statistics value (ratio of the departure of the estimated value of a parameter from its hypothesized value to its standard error). *P* Statistical significance value. FIV: VIF, Variance Inflation Factor.

The adjusted coefficient of determination, denoted by R2 adjusted, is 0.8274, which suggests that 82.74% of the total variation in the Y (house price/sqm) is explained by the relationships between Y and the selected variables. This implies that the model is significantly 'good fit' (Table 5).

**Table 5.** Goodness of fit of the model.

| S | R-Squared | R-Squared (Adjusted) | R-Squared (Pred) |
|---|---|---|---|
| 0.196382 | 85.34% | 82.74% | 78.75% |

The regressions results in the following:

Ln(Price €/m2) = 10,860 − 0.1346 × Ln(Rooms) + 0.2006 × Ln(OldCo) + 0.3811 × Ln(State) + 0.1446 × Ln(NeighSt) − 0.4057 × Ln(DistCBD) − 0.0639×Ln(DegpCo) + 0.0973 × Ln(ParkP) + 0.0439×Ln(MetroPI) + 0.0352×Ln(TrainPI) + 0.0683×Ln(IconPI) − 0.108 × Ln(PrsPI) + e

As stated by the result above, the regression analysis shows a statistically significant premium price for the location of the asset (in particular, location in the Central Business District) and for the parameter of dimensions, quality of the building, proximity of green areas, metro, and primary schools.

A statistical description of the independent variables intrinsic to each transaction is presented in the table below (Table 6).

**Table 6.** Statistical description of the residential intrinsic independent variables.

| | PRICE [€/M2] | Rooms | Old Coefficient | State | Status of the Neighbour-hood | Distance to the Center [m] |
|---|---|---|---|---|---|---|
| Calculation Method | Given | Given | 1/10 × Years from construction year up to 1941 | 1 = bad 2 = normal 3 = good 4 = New | From 1=bad perception up to 6= Very Prestigious | Given |
| Interpretation | - | - | Lower = Recently built | | | - |
| Mean | 3.637 | 2.91 | 9.81 | 2.64 | 3.08 | 4.689 |
| Standard Error | 111 | 0.07 | 0.18 | 0.04 | 0.07 | 116 |
| Median | 2.980 | 3.00 | 9.90 | 3.00 | 3.00 | 4.373 |
| Mode | 2.200 | 2.00 | 9.90 | 3.00 | 2.00 | 1.872 |
| Standard Deviation | 2.083 | 1.34 | 3.35 | 0.66 | 1.38 | 2.171 |
| Coefficient of Variation | 0.70 | 0.45 | 0.34 | 0.22 | 0.46 | 0.50 |
| Kurtosis | 1.47 | 9.53 | −0.45 | 0.01 | −0.69 | −0.7 |
| Skewness | 0.67 | 2.34 | −0.03 | −0.27 | 0.35 | 0.2 |
| Minimum | 1.263 | 1.00 | 1.90 | 1.00 | 1.00 | 546 |
| Maximum | 12.432 | 10.00 | 15.70 | 4.00 | 6.00 | 9.687 |

As can be seen from the coefficients of variation (<1.0) of each intrinsic variable, the sample used manages to represent a general picture of all the property types that are generally transacted in the city during Q3 2019 without biases of location and/or physical characteristics; as can be seen furthermore, the skewness values allow to understand that the variables considered adequately represent the whole spectrum of property quality, year of construction, and location, represented by the distance to the city center. In the case of number of rooms, it is clear that the number of rooms in Milan's residential properties is considerably asymmetrical, with an average of three rooms.

In order to understand the distribution of the spatial variables that are statistically significant and therefore considered in the regression, a synthesis of the calculation method is presented in Table 7.

**Table 7.** Calculation method of each spatial variable.

| Index | Calculation Method |
|---|---|
| Degradation Proximity Coefficient | Σ of Area of degraded areas over distance to each specific transaction in km (at least 200 sqm within 2 km). |
| Big Park Proximity | Σ of Area of urban parks over distance to each specific transaction in km (at least 100 sqm within 3 km). |
| Metro Proximity | 10 × Σ of number of metro stations to the specified transactions over distance to each specific transaction in km (within 1.5 km). |
| Train Proximity | 10 × Σ of number of train stations close to the transaction over distance to each specific transaction in km (within 1 km). |
| Iconic Places Proximity | 10 × Σ of number of monuments and historical landmarks over distance to each specific transaction in km (within 1.5 km). |
| Primary Schools Proximity | Σ of number of primary schools over distance to each specific transaction in km (within 1 km). |

A statistical description of the independent spatial variable is presented in the table below (Table 8):

**Table 8.** Statistical description of the spatial independent variables.

| | Degradation Proximity Coefficient | Big Park Proximity Index | Metro Proximity Index | Train Proximity Index | Iconic Places Proximity Index | Primary Schools Proximity Index |
|---|---|---|---|---|---|---|
| Mean | 863.3 | 0.7 | 44.0 | 10.2 | 193.8 | 9.8 |
| Standard Error | 67.1 | 0.0 | 2.8 | 0.8 | 19.5 | 0.3 |
| Median | 259.3 | 0.6 | 33.9 | 9.0 | 35.4 | 9.9 |
| Mode | 0.0 | 0.6 | 0.0 | 7.2 | 0.0 | 0.0 |
| Standard Deviation | 125.0 | 0.5 | 25.3 | 15.0 | 365.8 | 6.3 |
| Coefficient of Variation | 0.48 | 0.83 | 0.75 | 1.67 | 10.32 | 0.64 |
| Kurtosis | 0.7 | 2.1 | 0.9 | 3.9 | 6,4 | 8.1 |
| Skewness | 1.1 | 1.7 | 1.3 | 1.8 | 2.6 | 1.9 |
| Minimum | 0.0 | 0.0 | 0.0 | 0.0 | 0,0 | 0.0 |
| Maximum | 5657.2 | 2.4 | 90.5 | 74.6 | 1681.3 | 42.3 |

As can be seen from the coefficients of variation, a strong aggregation of iconic places is evident, yet it has statistical significance, due to the fact that is not directly correlated with the spatial distance to the city center, and due to this aggregation, several transactions are not affected by iconic places. The remarkable city center public transportation connectivity is well represented, and yet the sample with transactions covering different neighborhoods is not affected by a particular aggregation of Metro stations in one single area.

The following map shows the general price situation of the used transactions. It is important to note that the map represents the data used to build up the model, and not the average price in the city's areas (Figure 6). It represents a snapshot of the prices per sqm for the 352 properties used to build the model; as can be seen, the transactions located in the city center and its business district (Above Brera area) presented at the time span of data collection (Q3 2019) the highest prices in the city. Some areas outside the historical center present a prime price, such as the Tortona area and Tre Torri.

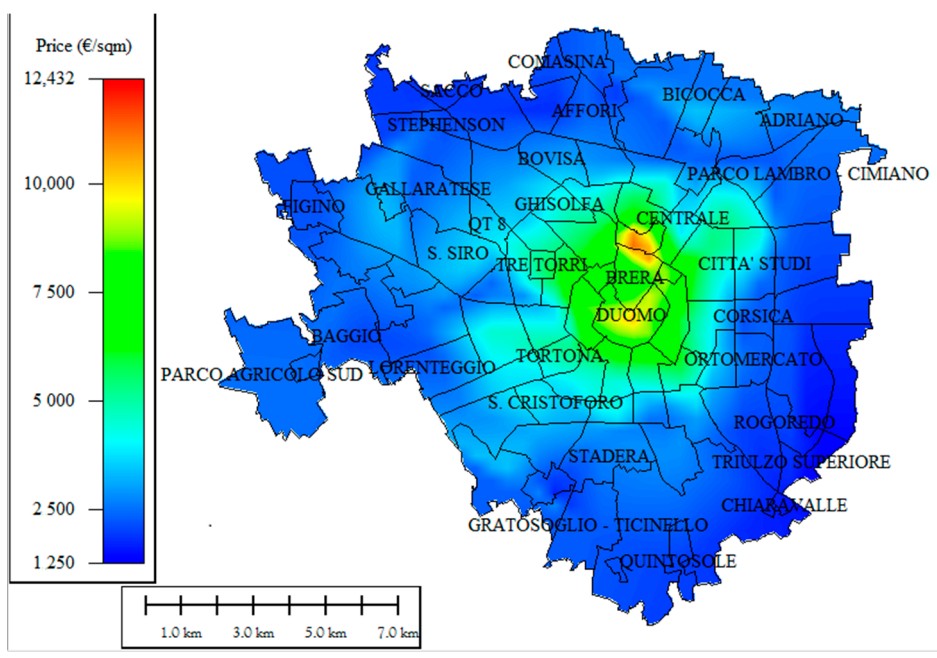

**Figure 6.** Prices of the transactions used for the model.

In order to geographically check the precision of the model, the resulting regression was used for recalculation the prices of the transactions used to build the model, and this

was directly compared the real data with the forecasted values. The following map shows the price per sqm obtained by running the model with the same 352 transactions (Figure 7).

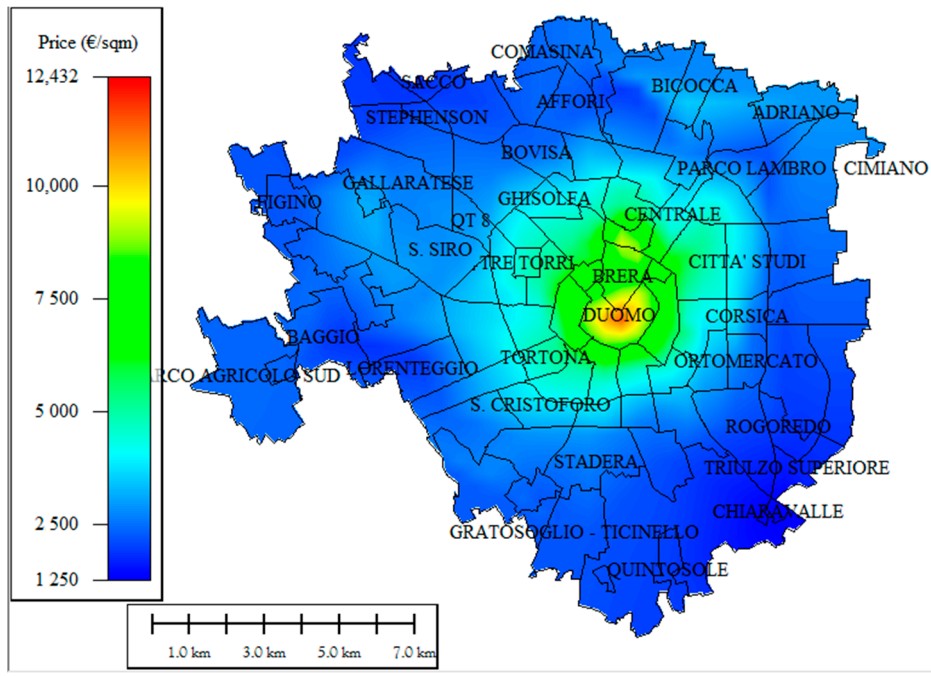

**Figure 7.** Prices forecasted by the model.

The similarity of Figures 6 and 7 can be highlighted by calculating the difference of prices per Sqm of the sample used to build the model and the forecasted price; Figure 8 shows the result of this calculation—areas in green are evidence that the model had a high precision, close to 0 €/sqm of difference between the forecast and the real price, and areas in red (es. Corsica) had a difference of circa 75 €/sqm, representing 2.3% of the price/sqm of the transactions in the area.

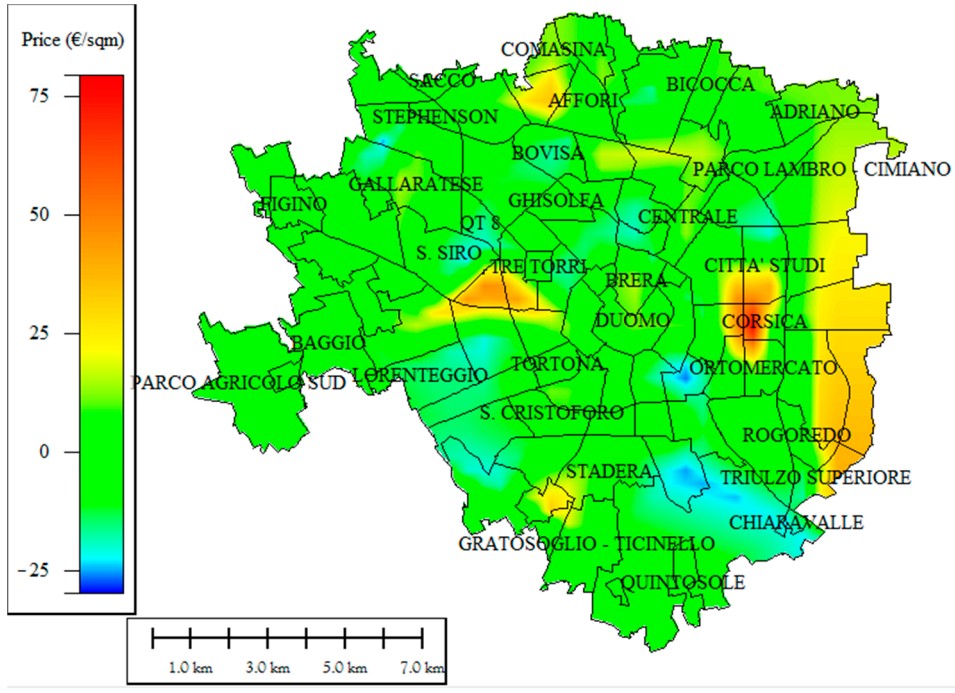

**Figure 8.** Difference between the forecasted price.

As can be seen, areas where the model failed in a percentage higher than a 5% difference investment analysis were not performed. The Corsica area gap is mainly due to abnormal transactions that took place in the area by the time of the transaction information gathering.

Results Interpretation: The house State (State), Old coefficient (OldCo), Status of neighborhood (NeighSt), and Big park proximity (ParkP) had the most significant positive impact on the housing prices, which were followed by proximity to Iconic places, metro stations, and trains. The relevance of the NeighSt coefficient implies that the people's preference for neighborhoods were weighted in a correct manner

The Distance to the city Center (DistCBD), number of Rooms (Rooms), Primary Schools proximity (PrsPI), and Degradation proximity (DegpCo) were found to have the most significant negative impact on the house prices.

The startling revelation of the results show negligible impact of Air quality, university proximity, contaminated area, and special green spaces on the housing prices.

The selected investment areas correspond to sectors where an interesting increase in price has been presented and where an attractive capitalization rate exists, besides the small difference (+−5%) between the model and the real data.

## 7. Conclusions

The forecasting model built in this paper revealed that the proximity to several amenities has a real impact on the price of houses in determined ranges, intrinsic characteristics and location have the greatest valorization power, but other local amenities such as parks, public transportation, and proximity to areas considered as degraded have a considerable impact on properties values.

The precision of the model was high. When analyzing the investment transactions difference between real asked price and forecasted values were on average close to 4%. When considering investment scenarios, the change of these factors must be considered as a long-term effect since, as demonstrated, there is a considerable delay in the rent anticipation of subsequent value changes within Milan's Metropolitan Area.

Several alternatives were considered in accordance with a scenario analysis in the property residential market. Different areas of the city were analyzed, specifically areas where the urban planning and redevelopment projects of the city are most interested in, always based on official information; not necessarily the areas where the current market proposals look, in general, already well established developed and modern areas. The analyzed scenarios change the results of the model for local areas in a deep way and allow an understanding of the future hot points of property value increases in the city, resulting in investment scenarios with IRR one-fifth higher than the average gross return of investment in the city.

The simulation of the different future scenarios for the city through the construction of a model that showed precision in the areas that were objective of the investment scenarios, allowing a proposal of different portfolios with investment characteristics and risk profiles that will certainly be more attractive that the average investment in other asset classes, reaching 205 bp over the average residential investment in the city.

Yet, the conditions of the Italian residential market present a considerable obstacle, as is the ownership distribution of the properties. Often having a single owner for each property, this situation was included in the investment scenarios by considering the risk connected with necessary bureaucracy and administrative proceedings. The analysis performed through the simulation of future scenarios permits reaching higher yields in properties located in areas with renovation projects that are certain to take place in the next 10 years. In this way, residential developments in these specific areas can overcome the resistance of foreign investors by offering yields that are higher than the average in semi-central locations that do not present high uncertainties in the feasibility of investments, neither in residential nor other asset classes.

The risk analysis was performed through analyzing the probability of the future interventions that would, as demonstrated, raise prices in the intervened upon areas, modifying the established capitalization rates that were obtained from last year's performance of the areas, reflecting the risks connected with refurbishments, administrative processes, and tenancy management. The total number of transactions reveals that the market still has room for greater demand; by 2021 it is expected to reach 2009 levels. The average increase in the analyzed areas in the last three years was +2.6%, yet it is still too early to predict if the current cycle will reach a peak and start decreasing in the next 10 years. This situation is reflected in the definition of capitalization rates for the investments that were analyzed. The proposed investments will remain very strong.

The analyzed areas have a considerable share of foreign inhabitants (Scalo di Porta Romana area with 22% for example). Considering that in the analyzed scenarios, besides the area trend of growth, the increase is going to be of 14.5% for Scalo di Porta Romana, a gentrification process should be further studied, since the percentage of inhabitants that own their house is close to 77,4%, so it may not involve a forced migration.

The analysis tried to represent the acquired data, in order to obtain a description of the real estate market trend of both the Municipality of Milan and the municipalities belonging to the Metropolitan City of Milan.

Then, we have drawn a trend over the period taken into consideration, looking for the relationship between changes in residential real estate values and changes in socio-economic factors.

In general, the trends recorded in OMI areas (Osservatorio del Mercato Immobiliare-Real Estate Market Observatory) from 2006 to 2017 for the residential sector were:

Central area: In the first period, 2006–2013, all the trends in the area were positive, with an 8% increase in the Centro Storico Duomo area, while capitalization rates were down throughout the period; in the second period, 2014–2017, the trends became negative, except for the Turati-Moscova area, which had an increase of 4.57%; in this period, the capitalization rate has stabilized.

Semi-central area: Central Station area had a positive trend in both periods with a total increase of 22%; it should be noted that only starting from the 2014–2017 period, Tabacchi Sarfatti area has had an important increase in price; it should also be noted that the zone associated with the Central Station area has been modified with the addition of more districts.

Peripheral area: In this case, both periods show a negative trend in prices for all areas, with a significant decrease in prices in the Lorenteggio and Forlanini areas.

Suburban area: In both cases, the trend is negative, with a sharp drop in the prices in the area corresponding to Gallaratese and Lampugnano.

Having then translated the values recorded by OMI into data relating to the 88 districts of the Municipality of Milan, the following values were drawn; however, the assumptions made to construct the model in order to translate the value associated with the zones into the value associated with the districts induces approximations in the extracted data:

- Tre Torri, with an absolute variation of + 147% (+13.43% per year), and an average annual increase of 0.47% without considering updating effects.
- Garibaldi, with an absolute variation of + 116% (+10.43% per year), and an average annual increase of 0.99% without updating effects.
- Portello, with an absolute variation of + 53.55% (+4.87% per year), and an average annual increase of 0.55% without updating effects.
- Washington, with an absolute variation of + 44.18% (+4.02% per year), and an average annual increase of 0.38% without updating effects.
- Isola, with an absolute variation of + 37.37% (+3.4% per year), and an average annual increase of 1.62% without updating effects.
- Centrale, with an absolute variation of + 17.76% (+1.61% per year), and an average annual increase of 1.39% without updating effects.

- Viale Monza, with an absolute variation of + 0.89% (+0.08% per year), and an average annual decrease of −1.24% without updating effects.
- Cascina Triulza Expo, with an absolute variation of + 16.31% (+1.48% per year), and an average annual decrease of −0.52% without updating effects.
- Lorenteggio, with an absolute variation of −12.4% (−1.33% per year), and an average annual decrease of −1.6% without updating effects.
- Trenno, with an absolute variation of −10.32% (−0.94% per year), and an average annual decrease of −1.42% without updating effects.

As for the analysis of the influence of some geographical, physical, and socio-economic factors, the state of preservation of the property is undoubtedly the factor with the most significant impact, followed by the quality of the neighborhood; the distance from the center is confirmed as an important factor of variation of quotations; the proximity to the city's large parks and the underground has an important positive effect on the prices. The change in status of the district, green area, and accessibility are the identified geographical factors that could signify a change in prices, given that the other spatial characteristics remain fixed. Other variation factors must be seen individually for each neighborhood (e.g., large real estate developments, market dynamics).

Moving on to the analysis of the 134 municipalities of the Metropolitan City of Milan, in the first period from 2006 to 2013 the most important total changes recorded are:

- San Donato Milanese, with an absolute variation of + 13.37% (+1.91% per year).
- Opera, with an absolute variation of + 14.21% (+2.03% per year).
- Locate di Triuzi, with an absolute change of + 10.7% (+1.53% per year).
- Trezzano sul Naviglio, with an absolute variation of −13.04% (−1.86% per year).
- Vimodrone, with an absolute variation of −10.02% (−1.43% per year).
- In the subsequent period from 2014 to 2017, the most important total changes recorded are:
- Pioltello, with an absolute variation of +8% (+3% per year).
- Vignate, with an absolute variation of +7% (+2.03% per year).
- Carugate, with an absolute variation of % (+2.01% per year).
- Buccinasco, with an absolute variation of −7% (−2% per year).
- Vimodrone, with an absolute variation of −5% (−1.4% per year).

It should be noted that even here, albeit less marked, there is an increase in residential real estate values with the proximity to the physical center of gravity of the Municipality of Milan. This highlights a strong attraction of the Municipality of Milan towards the other municipalities of the Metropolitan City.

However, following this analysis, the limitations given by the lack of a precise reference for transactions are evident, and hence, even if the database of Osservatorio del Mercato Immobiliare (Real Estate Market Observatory) aims to provide guidelines for all operators in the sector, it is necessary to study geo-localized data in one point to better study the phenomenon.

The effects of new real estate developments and services could in fact have a non-homogeneous effect for individual areas.

The importance of the identified individual demographic factors is linked to the redefinition of Piano di Governo del Territorio (PGT—Territory Governance Plan), which aims to accompany Milan towards 2030, focusing on various key factors:

- Reduction of economic and social imbalances.
- Integration with the urban region: The Metropolitan City offers great opportunities and different real estate scenarios to be taken into consideration; the heterogeneity of the municipalities is to be considered into a more connected city where, beyond the first rings, distance does not represent a differentiating agent of the prices and other property performance indicators.

- Improvement of environmental conditions: Reduction of pollution of greenhouse gases and carbon emissions; pollution has not been identified as a factor affecting the average price.
- Focus on each of the 88 districts: PGT proposes a differentiation between the 88 districts, and an individual plan for each district; it brings the suburbs close to the center through a system of squares (Loreto, Maciachini, Lotto, Romolo, Trento, Corvetto, Abbiategrasso) able to stimulate investments aimed at redesigning public space and promoting the renewal of neighborhoods. With regard to the importance of degraded buildings in lowering prices, Milano 2030 intends to stimulate regeneration processes of the degraded, vacant and disused buildings, especially public ones in working-class neighborhoods, through levers and incentives aimed at contrasting the abandonment of buildings and to facilitate renovations. The Plan introduces a new rule on abandoned buildings that severely penalizes those who do not redevelop or demolish (transferring existing rights) within a time determined by the PGT approval.

A full model built with all the factors considered in the Milano previsions, in addition to all the factors present in the studied model, would be an improvement for evaluating the general impact and global price behavior in the city, further, behavior of other similar HPM studies developed in the Italian reality could add different approaches in the inclusion of new variables.

**Author Contributions:** Conceptualization, M.M., G.C., and L.B.; methodology, M.M. and G.C.; validation, M.M., G.C., and L.B.; formal analysis, J.S.R.R.; investigation, J.S.R.R.; resources, J.S.R.R.; data curation, G.C. and J.S.R.R.; writing—original draft preparation, M.M.; writing—review and editing, M.M. and G.C.; visualization, G.C.; supervision, M.M.; project administration, M.M.; funding acquisition, M.M. All authors have read and agreed to the published version of the manuscript

**Funding:** This work was supported by the Real Estate Center (REC) of Politecnico di Milano and by the National Interest (PRIN) Project entitled "Metropolitan Cities: Economic-territorial strategies", carried out by Politecnico di Milano Dep ABC in partnership with the University of Bari "Aldo Moro", Naples "Federico II", and Venice "IUAV". REC incorporates the concepts of Academy, Lab and Research library; it is an incubator of wide-ranging initiatives in the real estate sector, with a strong vocation for scientific research, dialogue with operators and training.

**Institutional Review Board Statement:** Not applicable.

**Informed Consent Statement:** Not applicable.

**Data Availability Statement:** Not applicable.

**Acknowledgments:** The author would like to thank Gianni Guerrieri, Director of Observatory OMI, for the constructive cooperation and support. Real Estate Market Observatory (Italian OMI—Osservatorio del Mercato Immobiliare) is the principal official source reporting the national real estate market behaviour and, on the basis of a partnership deal between the Italian version of the IRS ("Agenzia delle Entrate") and Politecnico di Milano ABC Department, collaborated towards this project by providing us with a database.

**Conflicts of Interest:** The authors declare no conflict of interest.

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
