# Peer review of "Residential Property Behavior Forecasting in the Metropolitan City of Milan: Socio-Economic Characteristics as Drivers of Residential Market Value Trends"

_sustainability, doi:10.3390/su13073612_

Round 1

Reviewer 1 Report

This paper offers too much and finally deliver less. The research itself is a standard HPM investigation about explaining factors of Milan residential properties. The offered topics related urban structure, price predictions and others are not supported with the calculation. The article is not focused: many irrelevant elements presented in the manuscript, for example regional split of Milan and its legislation. Variables in the HPM model are not coherent, in Table 2 other variables and names are given than presented in the text. The sample of 392 properties is not described and analyzed. The transaction time of the sample is unclear. Referencing is not prudent and not unified. Some references are quite obsoleted. The language of the manuscript is unclear, careful editing and proofreading needed. Major revision recommended.

Author Response

Response to reviewer 1:

First of all thank you for your comments and suggestions.

To improve our work, we provided English revisions (these can be checked along the whole manuscript text) and we implemented the paper.

Each modification has been highlighted in yellow in the updated manuscript uploaded.

Reviewer 2 Report

My comments:

Paper is well done except the authors should consider a home's view when valuing it. I have no knowledge of the Milan market but if there are views from certain homes such as mountains or water or parks, they should be considered a component of value.
See:
 Pricing Residential Amenities: The Value of a View

* Earl D. Benson, * Julia L. Hansen, * Arthur L. Schwartz Jr. & *
 Greg T. Smersh /The Journal of Real Estate Finance and Economics/
https://link.springer.com/journal/11146> volume 16,  pages55–73(1998)

The Journal of Real Estate Finance and Economics Recent times have seen an expansion of theoretical and empirical research on real estate using the paradigms and...

 <https://link.springer.com/journal/11146>

Author Response

First of all thank you for the positive comment.

We have taken into account the suggestions.

Each modification has been highlighted in yellow in the updated manuscript uploaded.

Reviewer 3 Report

The study presents an interesting research, but the manuscript deserves careful review. The paper lacks in sufficient detail regarding the simulation studies and does not provide adequate explanation of discussion of the results in some cases.

ABSTRACT: This section should begin by presenting the context of the research, and not immediately present the objective of the study presented. Bibliographic references should be avoided in this section. Authors should add the results in this section (preferably numerical values).
KEYWORDS: Replace "Housing Price" with "Housing Prices Method (HPM)". This helps readers find the paper from search engines as SCOPUS. This section could be enriched with additional words.
INTRODUCTION: Quoting 6 references in a sentence without arguing is unprofessional and rigorous. Authors are requested to justify their choice of references included in the text (line 32).
Before the review section (line 33) authors should better argue the context of the research.
In the review section it would also be useful to mention the models used by the authors in the different studies (lines 33-59). Furthermore, it is not enough to list the studies, but it is necessary to argue and extrapolate some conclusions to support the study presented by the authors of this paper. Some parts are unclear, and the sentences incomplete or long (lines 94-95, 107-110). I ask the authors to read the text carefully and revise.
I suggest consulting other papers that have addressed the proposed theme by applying the method used by the authors: 10.1016/j.cities.2018.11.008, 10.1016/j.landurbplan.2008.06.005.
The authors starting from the analysis of the literature should highlight the lacks present in the literature, and explain the reasons that led to the development of this study. The objectives of the research are not very clear. Please the authors to clearly argue the research goals.
Add in a few lines the structure of the paper, describing what the different sections contain.

METHODOLOGY: In the manuscript there are two sections number 2.
Avoid writing a single subsection (2.1). This section may be structured differently. A first sub-section should present the general methodological framework. If possible the latter should be supported by a flow chart where to indicate the various steps, and the input and output variables.
This section describes the structure of the paper. But this description seems to be a structuring of the study framework, rather than the structure of the paper.
THE METROPOLITAN CITY OF MILAN (MCM) AS A PRIVILEGED STUDY CONTEXT and REAL ESTATE BEHAVIOR IN CITIES AND ITS METROPOLITAN AREA: These sections may be shortened. Enter only the information useful for carrying out the study.
ECONOMETRIC MODELING OF THE RESIDENTIAL PROPERTIES IN THE METROPOLITAN CITY OF MILAN: It is unclear whether the data from 2006 to 2013 will be used in this study. The authors first state that prices will not be used, but in the next section they propose an analysis over the 11 years.
Authors should read up on the hedonic pricing method. No references were made to Rosen's theory, and no references were made. The authors cite a work from 2009, while the method originates in the 70s.
It would be better to describe the variables in the table, indicating units of measurement and providing the description. Authors should also provide a descriptive analysis of the variables in the Appendix (min, max, standard deviation).
Tables 2 and 3 present the results of the regression. Authors should indicate what FIV, T value, COEFF means. EE. A correlation analysis would be necessary given the large number of variables entered. High FIV values ​​(which I believe stands for Variance Inflation Factor - VIF) should be compared with limit reference values. The performances of the first regression model should still be entered, even if the model contains non-significant variables.
Why did the authors choose a log-log model? The choice should be justified. Authors should explain how to interpret the results.
The results, however, are unclear. Authors should discuss and argue the results obtained.
It is not clear how the authors approach the problem of time in the regression model. I recommend the authors to read 10.1068/a45337, 10.1016/j.jclepro.2020.125327 and 10.1080/00036840701748995 where the issue of time in HPM models is addressed. Please explain better this question, declared fundamental by the authors in the introduction.
CONCLUSIONS: Authors should add future perspectives.

In general, adapt the bibliographic references to the journal template.Add more recent references (2020, 2021). I recommend that authors have the manuscript reviewed by a native English speaker.

Author Response

Thank you so much for your accurate and detailed comments and suggestions.

First of all to improve our work, we provided English revisions (these can be checked along the whole manuscript text).

We implemented the paper in according your suggestion to clarify and improve the different sections of the paper.

Each modification has been highlighted in yellow in the updated manuscript uploaded.

Round 2

Reviewer 1 Report

The quality of the paper was upgraded. However, some of my concerns were not addressed. The article still is not focused: many irrelevant elements presented in the manuscript, for example regional split of Milan and its legislation. The sample used by Authors is not statistically described and analyzed. The transaction time of the sample is unclear.

Results are described but figures are not interpreted.

 Within Keywords, "Housing Prices Method" mentioned as HPM.

Author Response

Dear Reviewer 1,

first of all thank you again for your time, comments and suggestions.

To improve our work, we added statistical description and analysis for intrinsic and spatial variables, added calculation method for non-given variables.

We added:

  • time span of data collection;
  • interpretation of Figures 5,6,7,8;
  • unified as Hedonic Price Method (HPM).

Each modification has been highlighted in green in the updated manuscript uploaded.

Reviewer 3 Report

I thank the authors for the changes made. Generally, a rebuttal letter is preferred for replying to reviewers, in which every single reviewer's comment is replied to, especially timely comments. 

Some typos are present. Please read carefully the manuscript text and figures (constant instead of "constant" for example).

The log-log transformation implies that the coefficients are a percentage change of the variable Y to a percentage change of the variable X.

A value-added to the results can be given by the comparison with other similar HPM studies developed in Milan. this does not weaken the work but validates the results obtained. It is not yet clear how to interpret the coefficients of the hedonic model. 

Author Response

Dear Reviewer 3,

first of all thank you again for your time, comments and suggestions.

As you suggested we read carefully the manuscript text and figures.

About the log-log transformation implies that the coefficients are a percentage change of the variable Y to a percentage change of the variable X, we noted and added in the model Box Cox selection description.

As you suggested about the value-added to the results we noted and added directly to conclusion.

Each modification has been highlighted in green in the updated manuscript uploaded.

Round 3

Reviewer 1 Report

After revisions it can be accepted